# Is the Viscoelastic Sheet for Slamming Impact Ready to Be Used on Glass Fiber Reinforced Plastic Planning Hull?

**Patrick Townsend** [1,*][ID]**, Juan Carlos Suárez Bermejo** [2]**, Paz Pinilla** [2][ID] **and Nadia Muñoz** [1][ID]

[1] ESPOL Polytechnic University, Escuela Superior Politécnica del Litoral, Facultad de Ingeniería Marítima y Ciencias del Mar (FIMCM), Guayaquil 09-01-5863, Ecuador; nmunoz@espol.edu.ec

[2] School of Naval Arquitecture and Marine Engineering (ETSIN), Universidad Politécnica de Madrid (UPM), 28040 Madrid, Spain; juancarlos.suarez@upm.es (J.C.S.B.); paz.pinilla@upm.es (P.P.)

[*] Correspondence: ptownsen@espol.edu.ec; Tel.: +593-991527396

**Abstract:** Planing hull vessel built with polymer matrix laminates and fiberglass reinforcements (GFRP) suffer structural damage due to the phenomenon of slamming during navigation, due to the impact of the boat hull on the free surface of the water at high speed. A modification in the manufacture of the laminates for these fast boats is proposed, consisting of the insertion of an additional layer of a hybrid material, formed by elastomer encapsulated in an ABS polymer cell. Using GFRP specimens made from pre-impregnated material and reproducing the characteristic impacts of slamming, it is possible to compare the modified material with the introduction of the viscoelastic layers with the response under the same conditions as the unmodified laminates. Additionally, the panels have been tested using impacts due to weight drop at different energies, which allow determining the material damage threshold as a function of the energy absorbed, and to establish a comparison with the GFRP panels modified by observation in fluorescent light. It is verified that the proposal to reduce the effect of these impacts on the generation of damage to the material and its progression throughout the service life of the vessel is effective.

**Keywords:** slamming; damage; viscoelastic layer; prepreg; OoA

## 1. Introduction

Slamming is an important event during the navigation of the ship, and it appears as a sudden force that vertically impacts the ship in the bow and generates energy from the impact between the hull and the water surface. This force translates into pressure that acts on a very small surface and is so unpredictable that it still requires investigation [1]. This impact and its damage to the ship is so important, that the sailors are very cautious, they reduce the speed so as not to suffer additional damage during the voyages. The complexity of the phenomenon is due to the fact that the fluid enters the bottom of the ship due to the angular difference between the body surface, expanding at high speed. This generates very high pressures that are very important issues in the design of the ships. The answers about this phenomenon and its influence on the structure of the vessels have not been fully resolved and it is more complicated when they are GFRP planning vessels [2,3].

Tests have been carried out with complete models that try to simulate the real scale of the effect of slamming on the boat [4]. Experiments with complete ship models look for the global answer, it is quite expensive and to this must be added computational models and long-term simulations that try to explain the damage caused by the pressure whip in the ship's material. This is directly related to their premature aging. For this reason, mathematical models and reproductions of the fatigue event on the ship's hull are an important option for analyzing the response of the impacted structure [5].

The material used in this work is that of ships constructed of GFRP based on pre-impregnation cured in the oven. The phenomenon of slamming for these materials has the particularity that the impact of the sea is converted into energy that dissipates, producing different levels of damage, being one of the most important parameters in the design of the ship [6]. The purpose of the investigation is to reduce the damage produced on composite material by slamming impacts, so that it does not expand within the laminate. And prevent it from skipping between the layers, causing intralaminar and interlaminar damages that change the flexural stiffness.

The use of viscoelastic materials has been an option to try to absorb noise and impact on structural surfaces. Its use in GFRP vessels has been studied in some ways in the laboratory, to observe its benefits in energy dissipation [7]. Protection against damage energy was successfully demonstrated as it takes advantage of the fact that composite materials have high stiffness ratios and moderate level of damping. Combining high levels of energy dissipation with minimal structural stiffness achieves good results [8].

Elastomers with their Poisson constant property close to 0.5 allow them great elongations and energy absorption when restrained, therefore they are the key to the design of the viscoelastic sheet. By encapsulating the elastomer within a rigid polymer, it allows to play with the weight of the sheet, its adhesion capacity and its thickness. Considering that it will be placed in the hull of the boats, the benefits of not increasing the weight of the ship is interesting. A hexagonal design is proposed, which allows creating a set of cells that are grouped together with each other when manufactured and can form an easy to apply set.

Taking advantage of the properties of viscoelastic, Hooke's Law in three dimensions shows that viscoelastic must have only one exposed and free surface for compression, and its other two restricted directions. In such a way that the proposed viscoelastic layer has exposed the elastomeric material to be compressed inside a capsule. As a rigid plastic, ABS was chosen in previous investigations, which is very common in the market. As a linear elastomer, the TPU type is also widely used today. The proposed analysis of comparing cyclical slamming pressures with vertical impact is directly related to the requested compression.

The viscoelastic properties of these materials, confined in a less deformable material, contribute in a very significant way to improve the performance of the material in service and dissipate a greater percentage of the energy received on impact, minimizing the damage generated in the material due to the slamming pressure peaks generated during navigation. For this reason, the energy produced in each of the impacts is studied in the laboratory from the perspective of the vertical impact due to drop weight, introducing an energy accumulated in the reproduction of slamming. This added to the microstructural observation, shows significantly the behavior of the damage evolution [9,10]

Among the questions pending in previous studies, was the problem of the adhesion of the viscoelastic sheet on the matrix. Through cohesive theory model studies, this concern has been answered for designers, who must consider within their structural calculations that the stress and strain limits do not exceed the adhesion and spread propagation thresholds.

The test performed is a modified version of the ISO 11343, a standardized Wedge Peel test method that allows to measure the dynamic resistance of structural adhesives to cleavage at different strain speeds. In the application that is envisaged for this viscoelastic layer in GFRP, designed for protection of the hull of high-speed crafts against slamming impact events, it is important to measure the adhesion strength between the protective layer and the laminate. The test used for this purpose consist of a wedge with an acute shape being driven at a defined velocity into the adhesive bond between both substrates. The wedge induces a bending causing the bond to fracture and adherends to peel apart. The cleavage force and the dynamic resistance to cleavage are consequently calculated from the test force-displacement curve. Because the method is basically grounded in the mechanics of bending beams, it can be analyzed quantitatively using standard fracture mechanics analysis.

The absorbed energy is considered as a representative parameter of the behavior of composite materials when subjected to impact loads, and to study the energy behavior during impact,

accelerometers with a computer data acquisition system have been used to quantify the energy returned. By observing impact forces and displacement, initial deformation and restitution are analyzed [11,12].

With the tests presented and the results are expected to answer the question designers if the viscoelastic sheet is ready for use in the construction of new boats GFRP. Energy parameters are also established to define how much the useful life of the planing hull vessel will improve, at the discretion of the manufacturers and users.

The present work is an extension of the previous investigations of the authors of this paper, referring to the use of the viscoelastic layer to improve the performance in the GFRP. The tests carried out in previous studies are presented in this article with complements that try to solve the new questions formulated by the designers: is it feasible to install the viscoelastic sheets in the GFRP planing vessel? Are they protected from the cyclical impact of slamming according to the previously resolved benefits and shortcomings? [13,14] For this reason, the cyclical slamming tests are complemented with the vertical impact due to weight drop to present its benefits from the perspective that the designer wishes to observe. The research aims to show another type of mechanical response of the viscoelastic layer under the perspective of vertical impact, and that the builders have different options for its use. The sheets are inexpensive and easy to cure in the construction of planing hulls.

The work has been organized according to the flow of activities shown in Figure 1, in order to present the experimental results from the definition of the experimental method, the slamming pressure levels to be used in vertical impacts, and with this information present discussion in which the comparison of results shown.

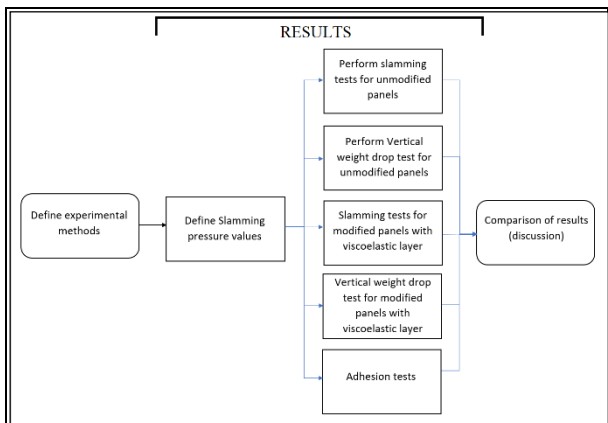

**Figure 1.** Organization of the research presented.

## 2. Materials and Methods

The reproductions of slamming impact were made with the equipment presented in Figure 2. It is composed of a variator-electric motor assembly coupled to a cam with a shaft supported by two flexible supports to dissipate the reaction energy of the cam when pressing the panel and protect the drive from possible excessive lateral loads. The cam is keyed while remaining fixed to the shaft and rotates between 200 to 320 RPM. The panel was installed at its base with gauge screws that allowed exerting pressures between 0 to 1200 $kN/mm^2$ and the panel is adjusted crosswise to the cam. A revolution counter was installed, and the temperature control was carried out with a thermal imager, to avoid that the temperature does not reach the glass transition temperature of the material due to friction.

Two types of panels were made of GFRP, without viscoelastic layer which was called "unmodified" and with viscoelastic layer which was called "modified". Some were made from Gurit WE-91 triaxial (0°/45°/90°) pre-impregnated OoA (Out-of-Autoclave curing), cooled to −18 °C, vacuum cured. The panels were manufactured using 3 fabrics of 1 mm each, cut to 270 × 270 mm corresponding to the

frame dimension of the test equipment. In the panels with the viscoelastic sheet, it was included after the first triaxial sheet. All the panels included a strain gauge at 40 mm from the impact zone.

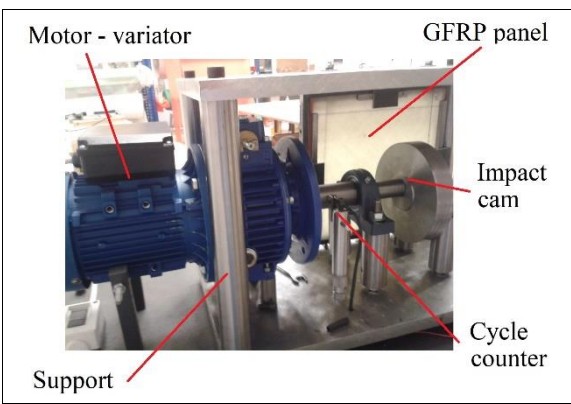

**Figure 2.** Reproduction equipment for slamming impacts.

For the manufacture of the viscoelastic sheets, a printer was used with two independent extruders so that there is no contamination between the materials. They were printed on a thermal surface. For the manufacture of the outer capsule, 3 mm acrylonitrile butadiene styrene or ABS was used. A linear thermoplastic polyurethane or TPU was used for the inner elastomer. Figure 3 shows a generic viscoelastic layer. Depending on the manufacturing date for the experiments, the color type of the ABS varies, so there are black and blue.

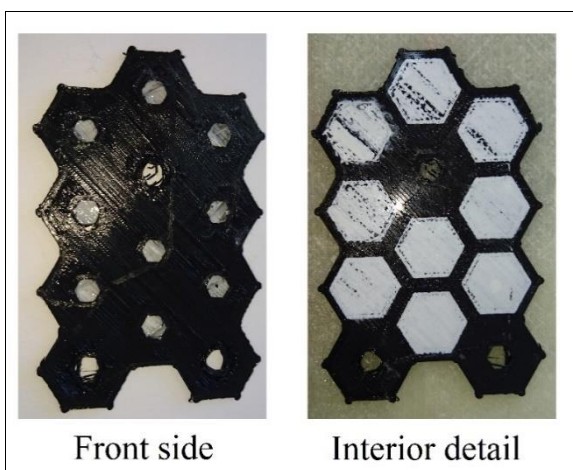

**Figure 3.** Generic viscoelastic layer.

The curing process of the panels was with a vacuum pressure bag and in an oven at a temperature of 120 degrees Celsius for 90 min. This process was applied to unmodified panels as well as modified panels.

A 12-m-long vessel constructed from GFRP was selected, and the maximum allowable slamming pressure values were calculated. This pressure occurs from the midsection towards the bow at the bottom of the ship. The purpose was to perform the slamming reproductions with lower values, in such a way that the failure of the material is due to propagation of the damage due to the cyclic load. The ABS classification rules for high-speed boats, Chapter 3, Section 2.2 [13] were used for the calculation. The ship's bottom pressures were calculated for the fully loaded condition, operating at maximum speed according to the selected geographic navigation area: The Galapagos Islands.

Three-point bending tests were performed to experimentally determine the pressure threshold from which damage begins to occur on the unmodified material. Cured specimens were cut from a panel into strips 15 mm wide by 250 mm long. They were flexed beyond their elastic limit until the

tension machine registers non-linearity, indicating the appearance of the first micro-cracks equivalent to the damage threshold as seen in Figure 4.

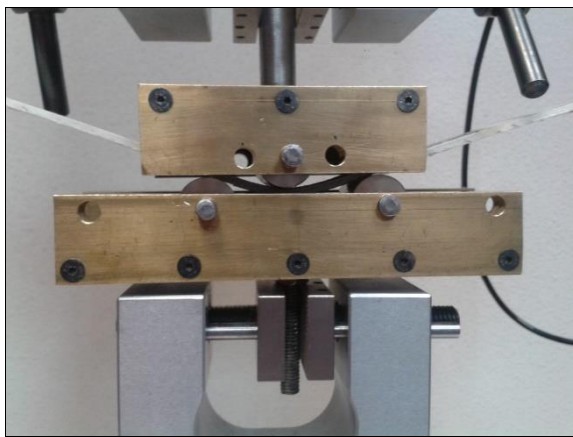

**Figure 4.** Three-point Bending test.

A machine for vertical weight drop testing was built. The impactor carriage falls by gravity on two chrome rails to reduce the effect of friction. The panels to impact were installed at its base, with all its edges embedded. It has an anti-bounce system with a laser reader to control the number of impacts. It has an acceleration sensor or gravitometer installed that sends the information to a data acquisition system and tabulates the acceleration G-force versus the time of impact. The equipment has a frame-shaped structure to which all parts are secured. The impactor was made with electromagnets for holding and launching. The guide bolts allow more weight to be added to it. At the bottom of the impactor it has the impact tip with a magnetized sphere. The number of impacts is controlled with a laser reader. Figure 5 shows the detail of the impact equipment and its parts.

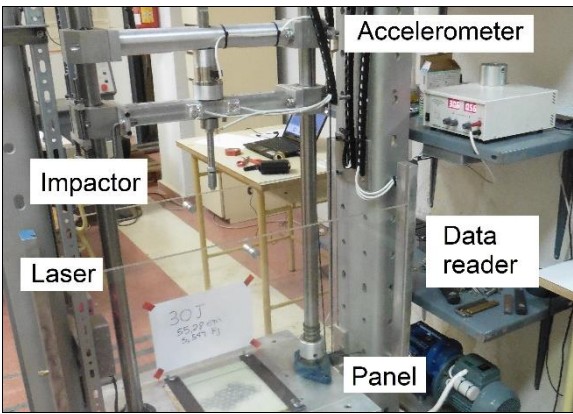

**Figure 5.** Weight drop impact equipment.

With the data on the forces of gravity read by the accelerometer, the following formulation is applied to calculate the energy absorbed and energy returned.

$G_i$: dimensionless number of gravities $i$ given by the gravitometer.

For each $Gi$ data reading the following parameters were calculated:

$\Delta t_i$: time interval (s)

$M$: impactor mass (kg)

$f_i$: impact force (N)

$x$: impactor displacement (m)

$k_i$: kinetic energy of impact (J) received by the impactor

$E_i$: impact energy (J) delivered by the impactor

$E_o$: initial energy of the impactor (J)

$E_{a\ i}$: energy absorbed by the material (J).

The results of the impact tests were processed with the following formulation to obtain the values under the curves of impacted energy and absorbed energy, and to be able to estimate the energy that transforms into damage.

$$f(t) = 9.81 * \text{M} * G_i \tag{1}$$

$$x_i = \frac{3}{2} * 9.81 * G_i * \Delta t_i^2 \tag{2}$$

$$k_i = \frac{1}{2} * 9.81^2 * G_i * M * \Delta t_i^2 \tag{3}$$

$$E_i = \frac{3}{2} * 9.81^2 * G_i * M * \Delta t_i^2 \tag{4}$$

$$E_{a\ i} = E_o - (|E_i| - |k_i|) \tag{5}$$

The impact tests were carried out in different energy ranges, varying the impactor weight and height.

To observe micro cracks within the laminate after impact and to make a comparison between unmodified and modified panels, characterization was performed with fluorescent penetrating inks to expose the sections to ultraviolet light with the procedure presented in Figure 6. The panels in the impact area were cut from 60 × 60 mm and drilled with a 0.5 mm drill bit, for the purpose of immersing them in a fluorescent penetrating liquid. Sectional cuts were made in the specimens, to observe their different intralaminar delaminations in interlaminar with fluorescent light. In the case of panels with viscoelastic sheet, this was withdrawn during exposure, because it is important to compare the viscoelastic layers protected.

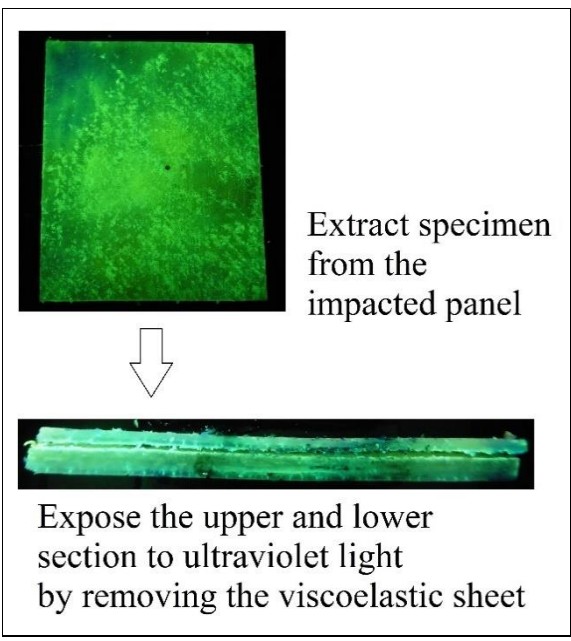

**Figure 6.** Test evaluation procedure for fluorescent penetrating inks.

In order to know the adhesion force that the viscoelastic sheet has once cured in the modified panels, shear tests adjusted to the Cohesive model theory were performed to estimate the damage initiation phase force, the maximum force at which detaches the sheet and the behavior of the propagation force of the detachment. For the shear tests, single-layer panels of Mat 200 were manufactured. This allows

the blade to work only on the viscoelastic sheet and the matrix as seen in Figure 7, the universal testing machine was used with a very fine steel blade, which will make the cut parallel to the viscoelastic.

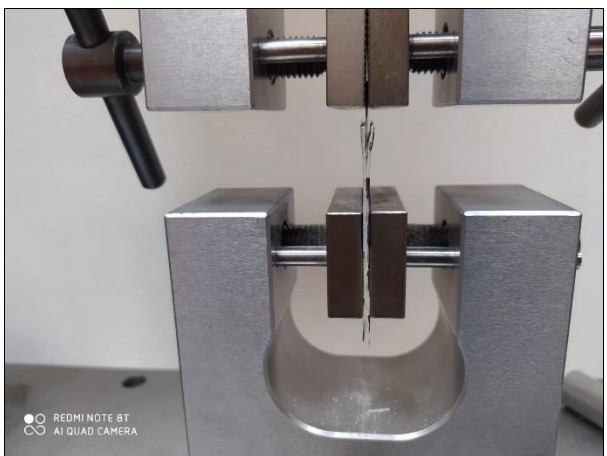

**Figure 7.** Shear test equipment.

## 3. Results

### 3.1. Slamming Pressure Values

For the calculation of slamming pressures according to the ABS classification rules, the typical GFRP vessel with length between 12 to 15 m and sailing at a nominal speed of 22 knots was considered. The site of operation is in the Galapagos Islands with an ocean of Beaufort scale 4, that is to say with waves of height of 1.5 to 1.8 m and frequencies of 14 to 19 s of bottom sea. The maximum design pressure at the bottom of the boat was between 1800 kN/m$^2$ to 1050 kN/m$^2$.

From the results of the three-point bending test to determine the damage threshold of the laminate, it was determined that over the 2300 μm/m of deformation, the curve was no longer linear and corresponded to the desired threshold.

Using the Finite Element Method (FEM), the force that caused deformations equivalent to those that cause damage to the material was estimated in the model. Figure 8 shows the symmetric model with the load at its center and a detail of the section in the impact zone of the same model. The results of the elastic strain have been plotted for a value in the maximum impact zone of 834 μm/m corresponding to a pressure of 385 kN/m$^2$.

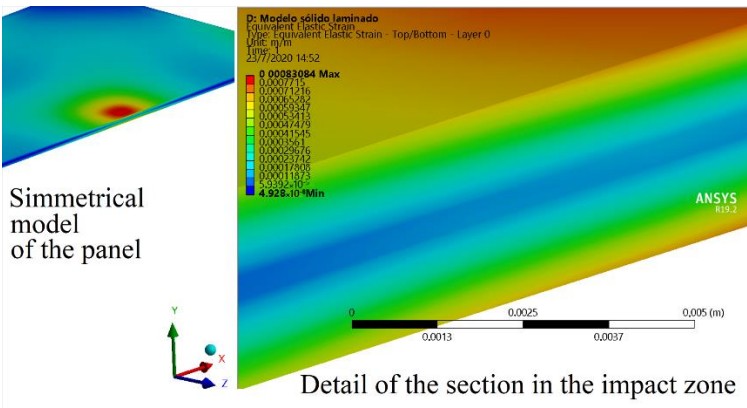

**Figure 8.** FEM model.

The pressure range in which the slamming tests could be carried out was determined to be 260 to 820 kN/m². The same procedure was applied to calculate the damage threshold pressure, which was obtained from 358 kN/m².

### 3.2. Slamming Tests for Unmodified Panels

Table 1 shows the tests carried out on the unmodified panels. The variables presented are the slamming pressure in kN/m² applied by the equipment on the contact surface, the number of cycles applied and the frequency in RPM of the impacts. The frequency variation depended on the motor-variator, so that the pressure on the cam does not produce loads that affect the system.

**Table 1.** Variables used in unmodified panels for slamming tests.

| Panel # | Pressure (kN/m²) | Cycles (×10⁵) | Frequency (RPM) |
|---------|------------------|---------------|-----------------|
| A | 260 | 2.10 | 210 |
| B | 400 | 1.50 | 220 |
| C | 630 | 1.81 | 220 |
| D | 810 | 0.21 | 310 |
| E | 830 | 0.21 | 310 |
| F | 420 | 0.27 | 310 |

In the 810 kN/m² and 830 kN/m² panels, the number of cycles of each group in these panels depended on the temperature of the cam which, due to friction, reached 70 °C. These panels had greater damage, in a low number of cycles of the order of $2 \times 10^4$.

Panel D reached 85% damage in the area of contact with the cam. It was observed that the first micro cracks appeared at $2 \times 10^2$ impacts and were already highly visible at $1 \times 10^2$ cycles. The first damage observed as slight white shadows were in the areas where the sides of the cam meet. They then lined up toward the center of the contact surface and show the evolution of damage as slamming strokes increase. Figure 9 shows the results of the test.

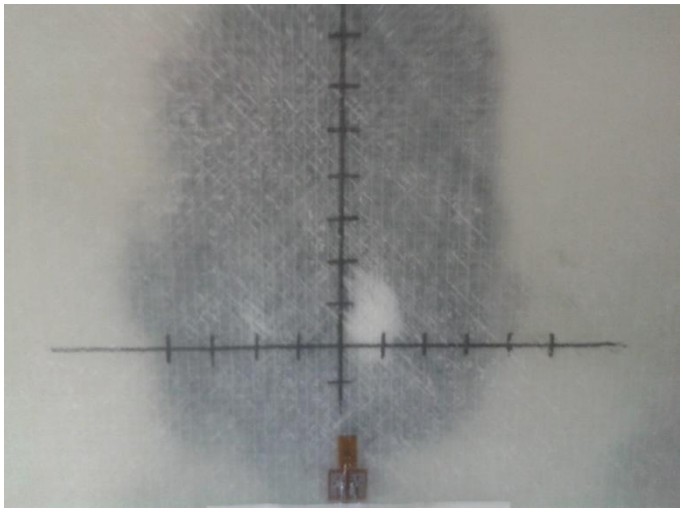

**Figure 9.** Evolution of unmodified panel D damage at $0.21 \times 10^5$ impact cycles with P = 810 kN/m².

Damage spread could be observed in real time at this pressure level and was first expanded by the penultimate laminated −45° layer on the strain gage side. At the end of the test it was found that breaks had already formed in the panel that passed from side to side in the laminate. Normalizing the percentage of damage between the maximum damage area value for each test in panels A–E. It is observed in Figure 10 that the evolution of the damage has a similar behavior for low energy impacts,

and similar behavior for high energy impacts. This according to the pressure values presented in Table 1 values used in the slamming tests of unmodified panels.

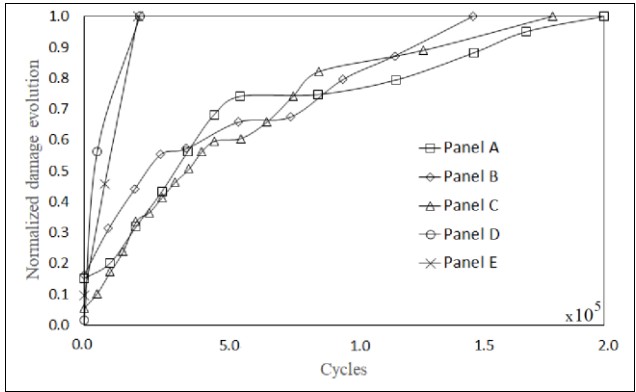

**Figure 10.** Normalized damage evolution on unmodified panels.

The change in flexural stiffness was evaluated with the results of the strain gage using panel F. In Figure 11 are the results of the for a deformation of $1 \times 10^3$ μm/m at the slamming pressure point in the $0.27 \times 10^5$ test cycles with a pressure of 420 kN/m². The data has a representative ascending line, which indicates that as the micro cracks increase, the panel loses rigidity. This is in accordance with the type of damage observed on the panel on the opposite side of the impact. These values were calculated from the deformations obtained with the strain gages.

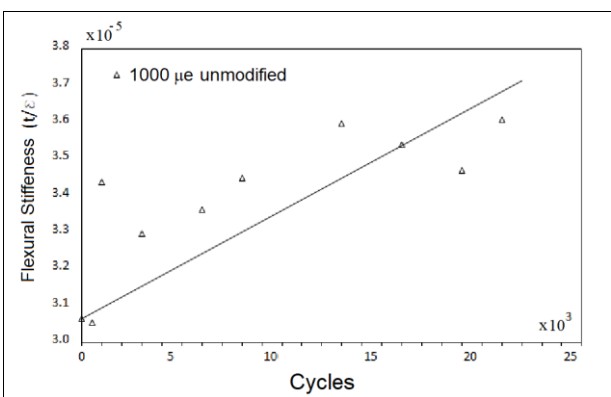

**Figure 11.** Evolution of flexural stiffness in panel unmodified F up to $0.27 \times 10^5$ impact cycles with P = 420 kN/m².

These results will be considered for the relationship between low energy slamming impacts and vertical weight drop tests. It is clearly identified in the results that some values presented in the graph correspond to high energy tests, and other tests correspond to low energy tests. Composite material has a defined energy absorption boundary.

### 3.3. Vertical Weight Drop Tests for Unmodified Panels

The impact tests were carried out in different energy ranges for a certain number of rebounds, and with the strain gage the maximum deformation of the panel was measured. These results are shown in Table 2.

Figure 12 shows the sequence of damage to the panels on the impact side, for the different energies tested with vertical weight drop. The sequence allows visualizing the increase in damage that occurs as the applied energy increases.

**Table 2.** Variables used in unmodified panels for vertical weight drop tests.

| Impact (J) | Maximum Deformation (μm/m) | # of Rebounds |
|------------|-----------------------------|----------------|
| 10 | 502 | 1 |
| 20 | 749 | 1 |
| 30 | 1130 | 1 |
| 40 | 1401 | 1 |
| 50 | 1710 | 1 |
| 60 | 1828 | 1 |

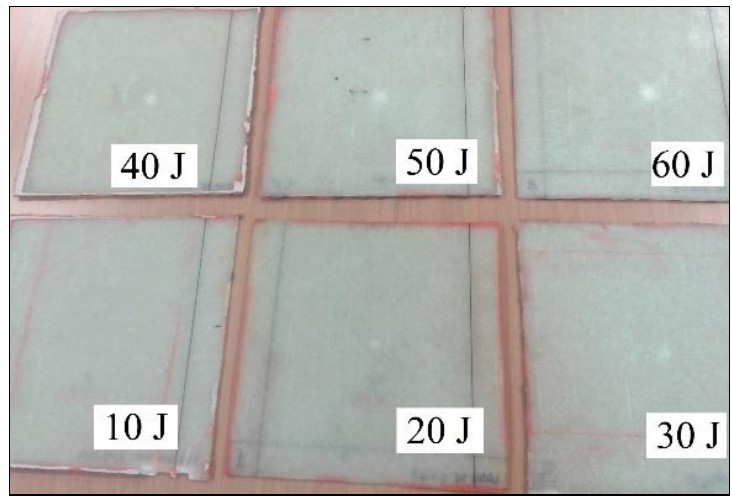

**Figure 12.** Unmodified panels impacted by vertical weight drop test from 10 J to 60 J of energy.

As the impact energy was increased, the microcracks were observed to be oriented in the direction of the fiber. The largest number of observed micro cracks evolved over the 45° layer. To a lesser extent in the 0° layer. Impacts less than 40 Joules can be considered low energy, while those greater than 40 Joules high energy, where the panels already show delamination and deep separation of layers. The phenomenon of slamming is a low energy process, for which the panels impacted at 30 J are the best to be considered for comparison with the panels tested in fatigue.

When characterizing the panels tested with penetrating inks, it was observed that they had indeed entered the delamination, being easily seen under ultraviolet light. Sections sequentially are exposed to fluorescent light as seen in Figure 13 for impact at 40 Joules respectively, in which interlaminar delamination can be easily observed.

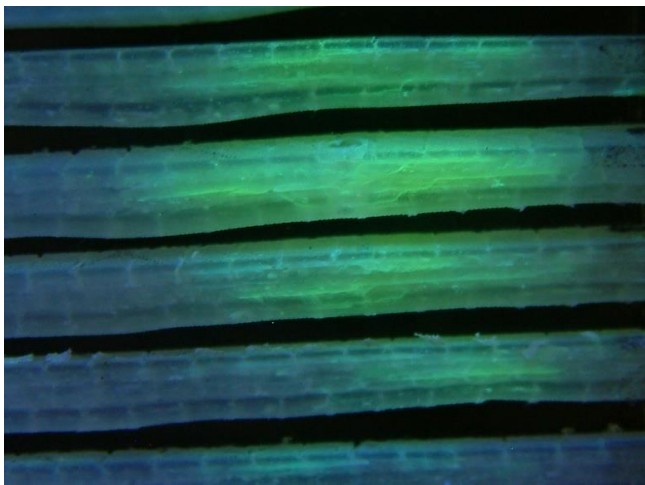

**Figure 13.** Unmodified panel sections exposed to ultraviolet light impacted at 40 J.

As expected in a laminated composite material, the breaks in the matrix are vertical and connect with delamination between layers, producing a ladder shape. The breaks observed in the lower layers indicate that the impact energy was converted into damage and was jumping from one layer to another. It is also important to note that according to the orientation of the layers, the delamination moves from left to right starting from the vertical direction of impact.

The accelerometer records the acceleration force (G), at a frequency of 104 Hz. According to the suggested equations, they show that the acting force versus displacement has the profiles presented in Figures 14 and 15. The peaks of damage that occurs as the impactor breaks each of the layers. The impactor slows down each time a new delamination occurs. Records the acceleration variations when micro cracks occur in the matrix. In the case of the low energy impact of 30 Joules, the impact force reaches its maximum value by breaking layer 5. Thereafter the layers offer resistance to deformation. In the case of impact at 50 Joules of high energy, the impactor has enough force to continue breaking all the layers until 9 at which it has already bounced.

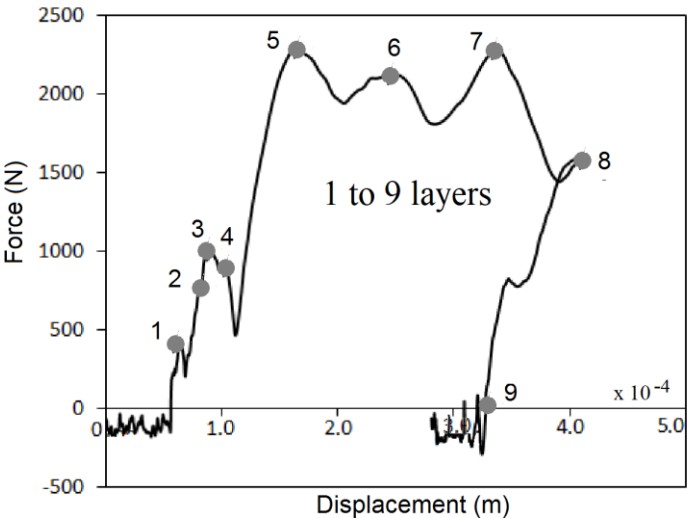

**Figure 14.** Comparison of the force vs. displacement diagram for an impact at 30 J in unmodified panel.

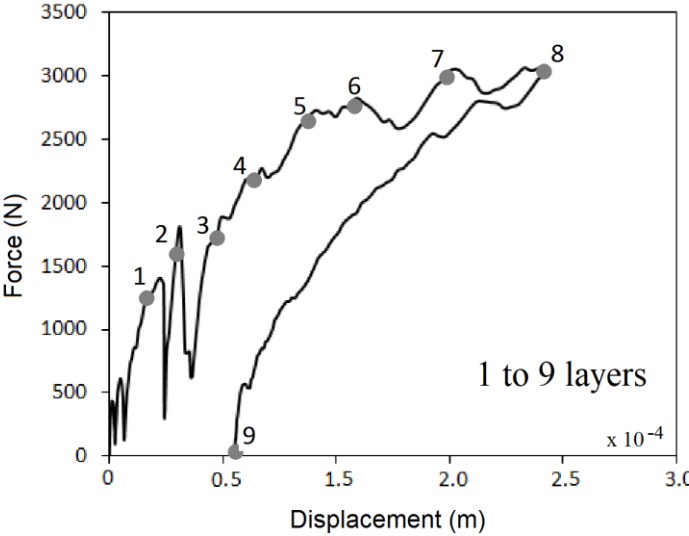

**Figure 15.** Comparison of the force vs. displacement diagram for an impact at 50 J in unmodified panel.

The acceleration curve also allows you to compare how fast laminate damage is and which layers absorb the most damage energy. In the case of the 30 Joule curve, between layers 4 and 5 there is a large increase in damage energy, indicating that this layer broke and delaminated. This is in accordance

with what was observed in the sections of the material exposed to penetrating ink and ultraviolet light. In contrast to the panel at 50 Joules, the high energy impact imposes a lot of kinetic speed, and the first layers up to 6 breaks at the same time interval.

The assessment of the absorbed energy is observed in Figure 16 for impacts of 20, 30 and 40 Joules. The curves clearly show that the panel absorbs energy that is returned in the form of kinetic energy, and over a certain time interval its capacity to absorb energy decreases, indicating that there is more damage within the panel.

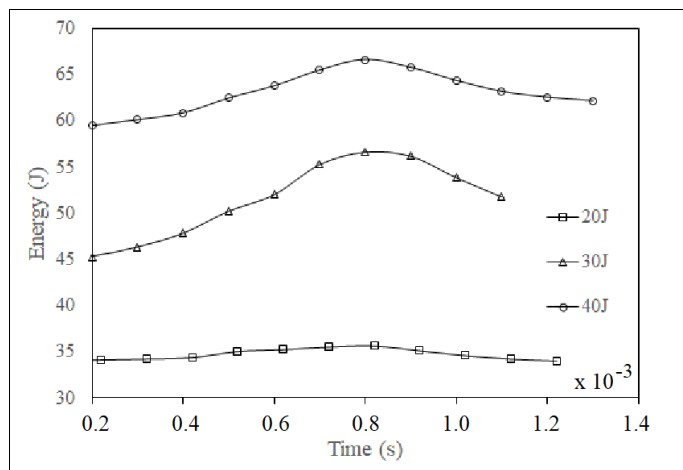

**Figure 16.** Energy absorbed by vertical energy impacts on unmodified panels.

The absorbed energy curve that has the best configuration is the 30 Joule curve, while the 60 Joule curve has an uncharacteristic trend, which is correctly related to the detachment and breakage of the matrix and the fiber that was observed in the test.

In the 20 Joule curve, its little variation is consistent with that observed in the panel after impact. The microcracks and directions of the delamination are very light. This indicates that little energy was transformed into damage and most was returned to the impactor.

*3.4. Slamming Tests for Modified Panels with Viscoelastic Layer*

The slamming tests carried out on viscoelastic modified panels are shown in Table 3. They correspond to the G panel test with high slamming energy whose microdeformation is close to the damage threshold. And the tests with panels H, K and L with cyclic impacts at low energy. The frequency variation depended on the motor-variator, so that the pressure on the cam does not produce loads that affect the system.

**Table 3.** Conditions of the slamming tests to modified panels with viscoelastic sheet.

| Panel # | Pressure (kN/m$^2$) | Cycles ($\times 10^5$) | Frequency (RPM) |
| --- | --- | --- | --- |
| G | 801 | 0.20 | 302 |
| H | 343 | 0.27 | 309 |
| K | 302 | 0.27 | 320 |
| L | 295 | 0.27 | 320 |

During panel G tests it was observed that over the $1 \times 10^3$ cycles, the first microcracks in the matrix were observed on the viscoelastic sheet. It was a high energy test and at the end of the test no delamination or cracks were observed on its surface. Panels K and L showed little damage to the matrix, and these were analyzed for adhesion with shear tests. Panel K is shown in Figure 17, which at $1 \times 10^3$ impact cycles already shows the appearance of microcracks, whose damage evolution was not significant.

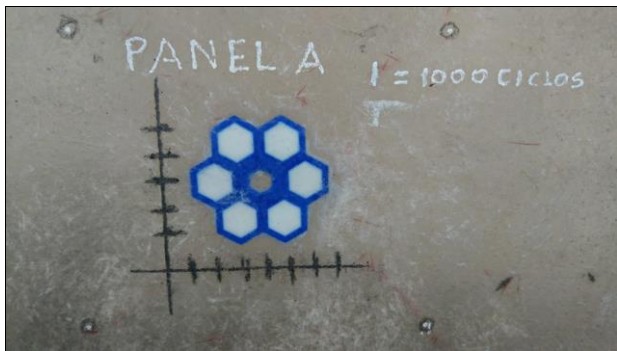

**Figure 17.** Modified panel K impacted at $1 \times 10^3$ cycles with 343 kN/m$^2$ slamming pressure.

The use of Mat fiberglass allowed the micro cracks that appeared inside the panel to be well visualized. During the test, the ambient temperature could be controlled so that there were no problems with the panel temperature. The control was made with the thermal imager and did not reach more than 50°. Also, the crystallinity of the resin, allowed to observe that there was no detachment of the viscoelastic during the test.

For the modified panel F, $2.7 \times 10^5$ impacts were made with a slamming pressure of 343 kN/m$^2$, evaluating its change in flexural stiffness. The change versus cycles is presented in Figure 18. It is observed that the representative line has a low magnitude slope and the data tends to change in a lower value. This is related to what was observed during the test, in which few microcracks and damage evolution after impact are observed. The protection of the viscoelastic layer is clearly observed because the damage is minimal on the face after impact. The orientation of fiber damage is not clearly defined. The results are considered good because they have an acceptable tendency, and in high energy levels there is also a concordance in flexural stiffness.

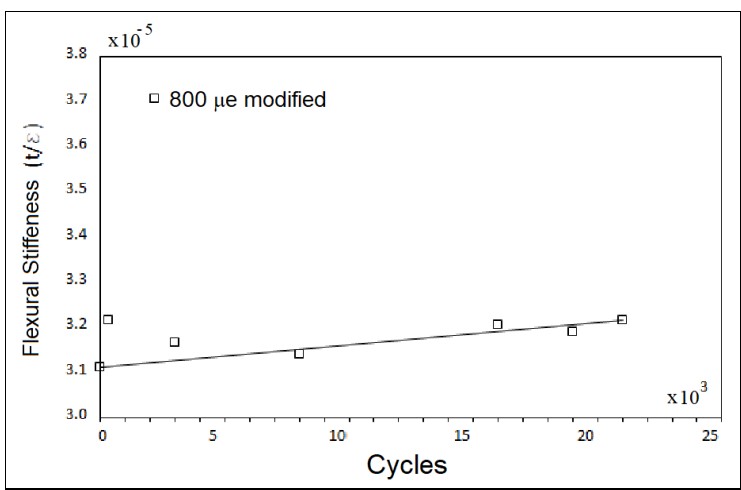

**Figure 18.** Evolution of flexural stiffness in modified panel F up to $0.27 \times 10^5$ impact cycles with P = 343 kN/m$^2$.

*3.5. Vertical Weight Drop Tests for Modified Panels with Viscoelastic Layer*

The tests for vertical drop in weight were carried out in different energy ranges, varying the impactor weight and height as seen in Table 4. This in order to observe results for low energy and high energy tests.

**Table 4.** Conditions of the vertical weight drop tests on panels modified with viscoelastic layer.

| Impact (J) | Deformation Maximum (μm/m) | # of Rebounds |
|---|---|---|
| 20 | 252 | 1 |
| 40 | 655 | 1 |
| 80 | 991 | 1 |
| 120 | 1040 | 3 |

The tests carried out on the panels with a viscoelastic layer with impact energies for weight drop of 20 and 40 Joules are shown in Figure 19. The impact of 20 Joules presents little damage on the surface with little failure of adhesion with the laminate matrix. In contrast, in the 40 Joules panel, the damage is much greater and there is a complete cradle separating the viscoelastic lamina with the matrix. The impact is punctual and there is a noticeable difference with the high energy slamming test. The vertical weight drop tests obtained similar results for the low energy case. Over 30 Joules the data obtained is applicable for a comparison.

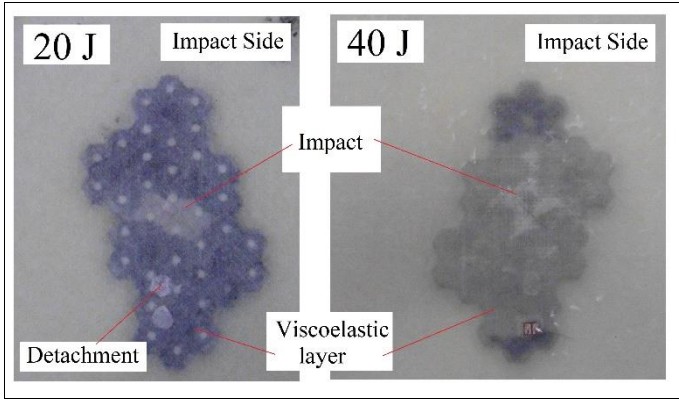

**Figure 19.** Impact tests for gravity weight drop of 30 and 40 J of energy to modified panels.

The modified panels were treated in the same way with fluorescent penetrating inks to assess the type of damage. During the cut, the viscoelastic sheet came off. This is seen in Figure 20. It should be noted that the viscoelastic layer came off due to the oil cut. The sections were 2 mm thick and despite the slow cutting speed, this affected their adherence. The cutting speed was very slow and oiled, to prevent fragments of composite material from sticking to the sections and showing erroneous results under fluorescent light.

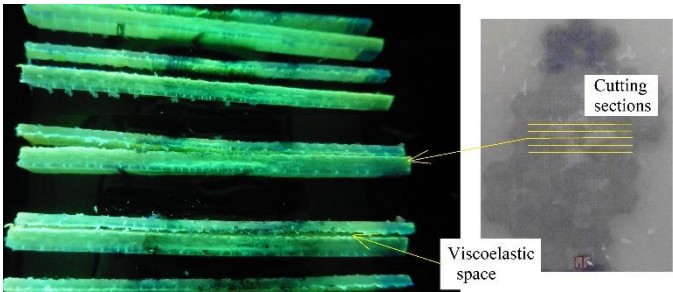

**Figure 20.** Characterization with penetrating ink to panels impacted at 40 J and 80 J.

It can be seen in the cross sections that the impacted laminate layers before the viscoelastic layer have damage to the matrix and fiber. The layers after this have little delamination. The matrix shows few vertical breaks and the ladder joints are not clearly visible. Delamination do not take a lateral direction due to the orientation of the layers but appear intermittently and hardly visible. All sections

exposed to ultraviolet light show the same behavior. Vertical impact is similar to a fatigue event shown in slamming trials.

According to the results of the accelerometer and the use of the respective formulation, the behavior of the panel during the impact at 40 and 80 Joules is observed in Figures 21 and 22. It is observed that the impactor breaks layers 1, 2 and 3, and then experiences a decrease in force. This means that the impact force is damped by the viscoelastic layer. The impactor breaks the next layers, from the decreased force causing less damage. Depending on the impact force, the impactor does more or less damage to layer 9 on which it has bounced. Both results, the presence of the viscoelastic sheet is fully identified. At the 80 Joules high-energy impact, the layers broke at similar time intervals, until the interference of the viscoelastic layer. The acceleration curve is clearly marked as it affects the impactor and protects subsequent layers.

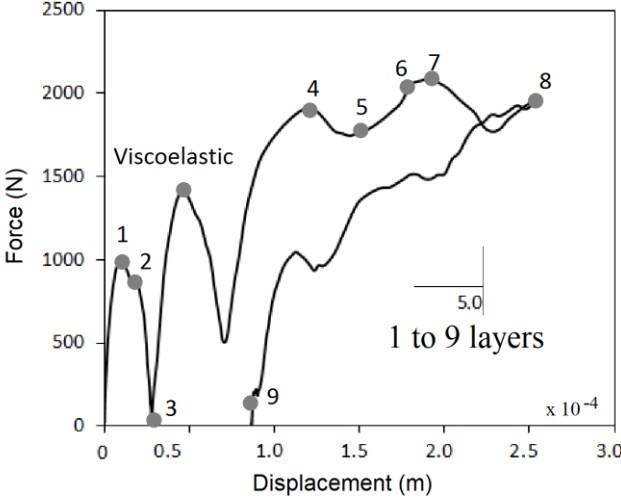

**Figure 21.** Comparison of the force vs. displacement diagram for an impact at 40 J in modified panel.

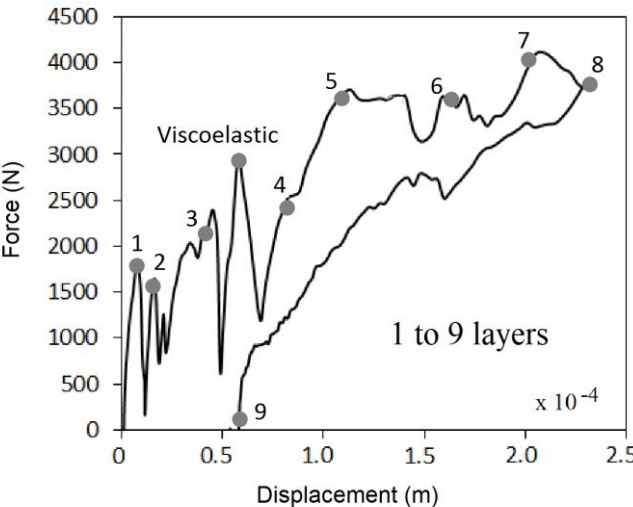

**Figure 22.** Comparison of the force vs. displacement diagram for an impact at 80 J in modified panel.

The energy absorbed by the modified panels for impacts of 30, 40, 80 and 120 Joules is shown in Figure 23. It is observed that the panels maintain their capacity to absorb impact energy, which indicates that the damage does not increase proportionally as which increases the impact energy. This is indicative of less damage to the matrix, which accumulates the kinetic energy imposed by the impactor. It is observed that at 120 Joules, the absorbed energy curve does not show the typical tendency of composite materials to high energy impacts. It does not have the ability to return all damage, but it does return

damage energy to the impactor. This is identified in the negative slope in the intervals that the impactor reaches layer number 9. In such a way that the curve clearly shows that the material receives little damage for high energy vertical impacts. This result is unrelated to slamming energy impacts.

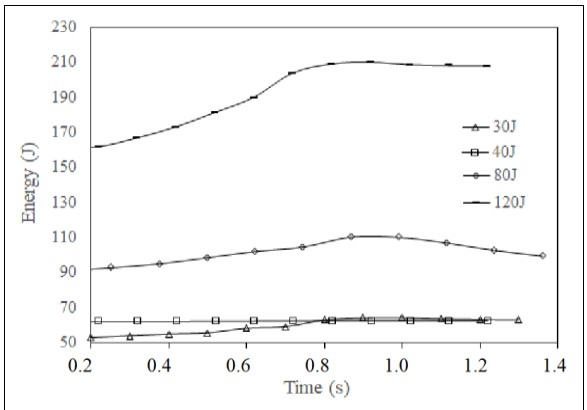

**Figure 23.** Energy absorbed by impact of weight drop tests on modified panels.

*3.6. Adhesion Tests*

Adhesion tests gave the expected results according to the "Cohesive Theory Model". Panels K and L were shear tested after being subjected to $0.27 \times 10^5$ slamming cycles. Figure 24 shows specimen K with the detachment of the viscoelastic. The tests were carried out on two sides in order to have more results to know the behavior of the adhesion of the sheet to the matrix.

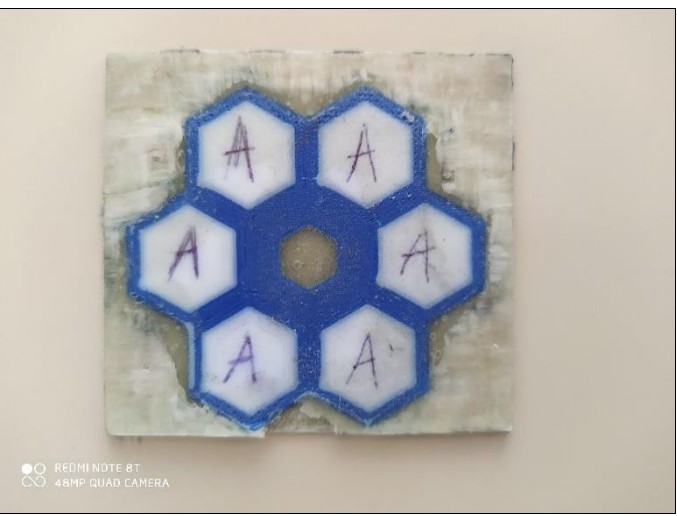

**Figure 24.** Panel K tested with the shear test.

Special care was taken when the specimen was placed in the presses, so that it is in line with the blade. The speed of the machine was calibrated so that the test is carried out at the minimum capacity. In such a way that the results are representative of the different acting forces in the cohesive zone

The resulting forces that were obtained in the tests are shown in Figure 25 for panels K and L. The damage initiation phase is clearly identified in each of the tests, and there is little difference in the value of maximum force. That is required for the viscoelastic sheet to peel off. Once detached, the team was able to test the propagation of the detachment of the layer. The maximum force value corresponds to the shear force of the laminate so that the adhesion fails.

The results of the test were bin visualized in the damage of the test piece, because the use of the mat cloth allowed it to be crystalline. The blade when entering between the viscoelastic sheet and the

matrix, could be followed to ensure that the test is well performed. The final data of the trial should correspond to the threshold of viscoelastic adhesion damage with an epoxy matrix.

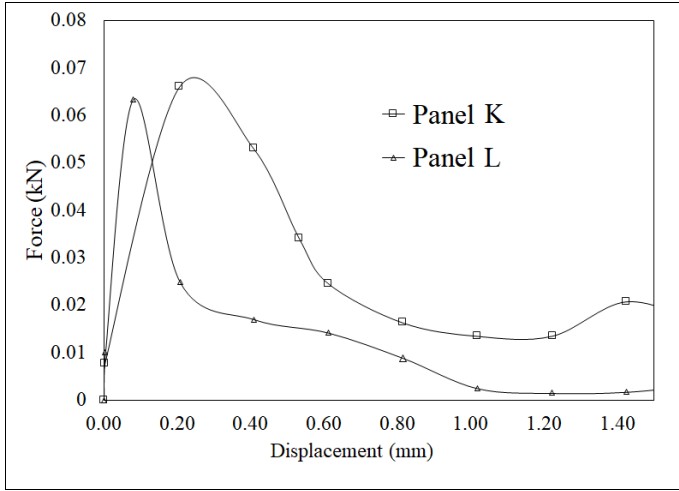

**Figure 25.** Adhesion force obtained with the shear test.

## 4. Discussion

If the damage caused by the impact of slamming that occurs in the hulls of the GFRP vessels could be observed in the tests. These do evaluate the evolution of the energy that is being converted into damage inside the laminate.

The insertion of a viscoelastic layer effectively mitigates the damage caused by the slamming phenomenon by reproducing the impacts under controlled laboratory conditions. The protection from the damage presented allows to absorb the dissipation of destructive energy, protecting the structure of the ship's hull and increasing its useful life. This brings a new perspective to the design of the ships and their scantlings, since the viscoelastic modification changes the way in which the stress concentrations are distributed in the hull of the boats. Comparing with the flexural stiffness values obtained along the scale of applied cycles, we observe in Figure 26 that over the $5 \times 10^3$, the unmodified and modified panels begin to distance in magnitude. The unmodified panel quickly gains flexural stiffness while the modified panel remains and does not change significantly.

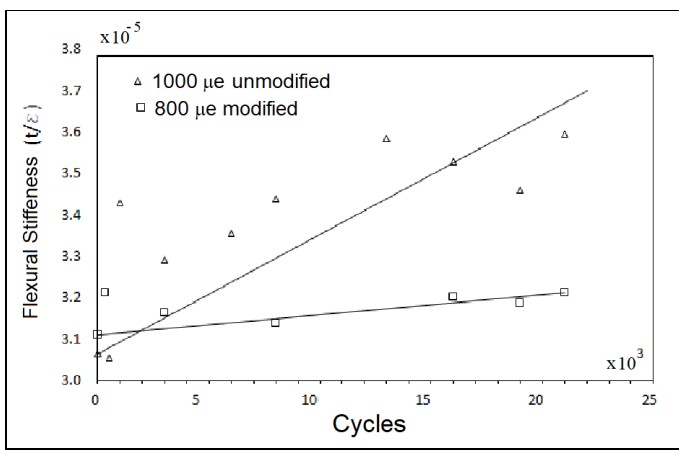

**Figure 26.** Comparison of the change in flexural stiffness during the slamming test with $0.27 \times 10^5$ cycles.

The panels impacted in pressure levels with values above and below the damage threshold calculated with the three-point flex test, marked an additional parameter to relate it to the maximum slamming pressure recommended by the ABS rules for the type of vessel selected.

Visual observation of the damaged surface within the contact area with the cam served to calculate the percentage of damage. This is related to the stress values that exceeded the damage threshold, transformed into microcracks.

The characterization by fluorescent penetrating inks, allowed us to physically observe how the damage does not propagate in the same way under the layers of the viscoelastic sheet, mixing this damage between the impact shock and the normal tension due to the flexion of the panel on the tensile side. Figure 27 shows two cut sections in the impact area already characterized by an unmodified and a modified panel, both impacted at 40 Joules. In the panel without modifying the interlaminar and intralaminar damages, they are linked, producing internal breaks in the laminate between layers 5 and 9 with important separations to the matrix. On the other hand, in the modified panel, the viscoelastic sheet protects the section of the laminate between layers 6 and 9, which correspond to those subsequent to the sheet. This means that it has returned the energy of damage and therefore little damage is observed on the underside, corresponding to microcracks and no delaminations.

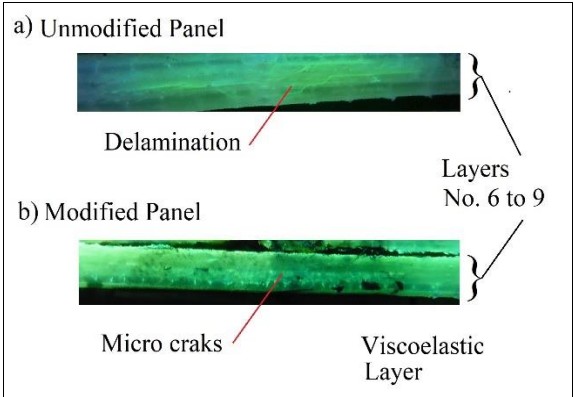

**Figure 27.** Comparison of the impact section for weight drop at 40 J characterized with fluorescent penetrating inks for unmodified panels (**a**) and modified panels (**b**).

Figure 28 shows the same comparison, but for a test of 80 Joules. Damage to the unmodified panel is very severe and unusable. On the other hand, the damage for the modified panel is less. If there is delamination of layers 6 to 9 that correspond to the layers after the viscoelastic layer, but not in all the layers. Viscoelastic layer protected these layers from severe impact damage.

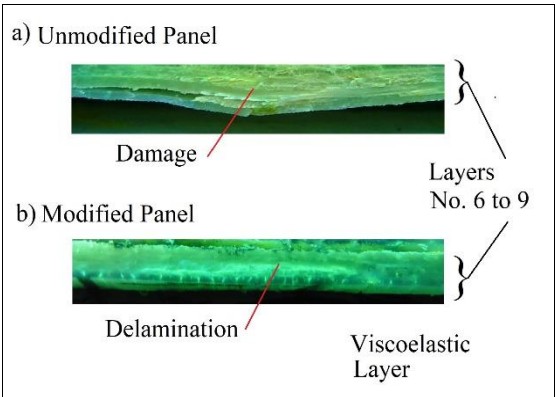

**Figure 28.** Comparison of the impact section for weight drop at 80 J characterized with fluorescent penetrating inks for unmodified panels (**a**) and modified panels (**b**).

According to these results, when comparing the absorbed energy calculations shown in Figure 29, there is a notable difference in the amount of energy for the same test at 30 Joules of impact. The modified panel absorbs little energy and returns it in a better way demonstrating protection.

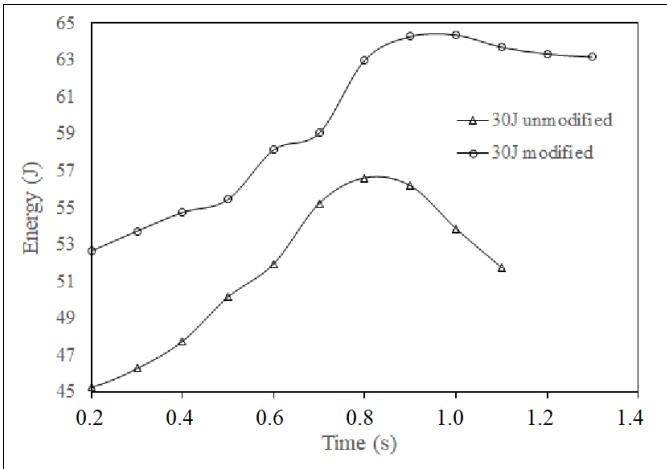

**Figure 29.** Comparison of energy absorbed by impacts of weight drop test.

## 5. Conclusions

The proposed methodology is a form of reproduction to assess damage during navigation, according to the slamming pressures that occur at the bottom of the hull of a boat. The pressures exerted on the GFRP for this case, is related to the percentage of damage produced per cycle with a tendency to increase pressure intensity. This is because the microcracks that align quickly with the fibers with the highest tension, are oriented to reach the fracture due to the decrease in stiffness and increased brittleness of the material. Penetrating ink tests confirm these results. The equipment is suitable for conducting low energy slamming impact tests in which the microcracks are aligned interlaminar for medium energy level impacts in which interlaminar damage also produces intralaminar damage. The impact energy was definitely absorbed by the viscoelastic layer. The viscoelastically modified GFRP protects itself from cyclical impact shocks from slamming, and the measure of maintaining the flexural stiffness is the ability to extend the life of the hull of ships made from this material. The adhesion values of the sheet allow designers to define their laminates so that the stresses of the material do not exceed the release values of the viscoelastic layer. The correct location of the viscoelastic sheets in the areas of greatest concentration of stresses due to the hit of the ship with the sea, is a starting point in the modification of shipbuilding to produce new types of more resistant ships. The viscoelastic sheet after this extensive investigation is ready to be inserted in the new constructions of planing hull vessels.

**Author Contributions:** In the research, P.T. carried out the execution of experiments, development of the formulations and narration of the manuscript. J.C.S.B. and P.P. collaborated with the conception and manufacture of the laboratory machines used in the research and the conceptual design of the experiments. N.M. with the conception of the GFRP application and revision of the manuscript. All authors have read and agreed to the published version of the manuscript.

**Funding:** This research received no external funding.

**Conflicts of Interest:** The authors declare no conflict of interest.

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
