# Peer review of "Is the Viscoelastic Sheet for Slamming Impact Ready to Be Used on Glass Fiber Reinforced Plastic Planning Hull?"

_applsci, doi:10.3390/app10186557_

Round 1

Reviewer 1 Report

The authors investigated the how the glass fiber reinforced plastic composite response to the impact test. Before the publication, I have some concerns before the publication:

  1. how the viscoelasticity is working in this composite? I suppose the impact generates heat and this heat can let the polymer turn to rubbery state form glassy state. This process and absorb a lot of energy. Please explain the mechanism in the introduction.
  2. In this study, the impact test was conducted by the dropping test. I suppose this is low velocity impact and can this velocity meet the real case happens for the boat?
  3. All polymers have viscoelasticity and why the ABS is chosen? Is this on purpose or just randomly find a polymer rubber?

Reviewer 2 Report

Dear Authors, 

The Reviewer found the paper interesting. The paper is written in an appropriate manner. The research objective is stated clearly. The research methods are described correctly. The results and conclusions are presented correctly. 

However, the paper requires a minor improvements. The Reviewer markted his comments and questions in the attached pdf file.

Please answer the questions which are marked on the yelow colour. The red marks are given for the Authors for possible improvements of the manuscript. 

Best Regards,

Reviewer  

Reviewer 3 Report

In this manuscript, the authors presented detailed experimental data in an effort to evaluate the feasibility of viscoelastically modified GFRP as a new material for the design of planning hulls. The data presented and conclusion drawn from this paper is interesting to the general audience in the area of composite materials.

The major concern with the manuscript is its novelty, which is not clearly explained in the manuscript. Indeed, the corresponding author has made significant effort in the same area of research, and has published experimental studies that assessed the performance of GFRP with viscoelastical layers under impact, see P. Townsend et al. Ocean Engineering 159 (2018) 253-267; P. Townsend et al. Key Engineering Materials 847 (2020) 3-8. It is surprising that the authors did not mentioned these papers in the Introduction section, as they are the most relevant studies to the submitted manuscript. In both of these papers, the authors concluded that the addition of a viscoelastic layer helped reduce the damage of slamming of composite materials. In their most recent study in Key Engineering Materials, the authors already concluded that “the proposed modification is the future of shipbuilding of this type of vessel”. Therefore, the natural question is: what new information does the current manuscript add to the knowledge already published before? 

Here are some specific comments related to the details of the manuscript:

- The authors should proofread the manuscript. Some typos, such as missing/redundant ‘.’, repeating words, and wrong numbering of sections, can be easily avoided.

- The writing of the manuscript needs to be improved. Specifically, writing in the Results section is fragmented, with paragraphs consisting of only 1-2 sentences. The flow of the presentation should be improved.

- Data presented in Figs 25-29, fits better the Results section (Sec 3).

- Size and resolution of Fig 4 is too low to show details of the experimental setup.

- Colorbars should be added to Fig 7. Do the two subfigs share the same colorbars?

- Sentences on line 287 and 289 are confusing.

- Fig 19: At what angle are we looking at the panels? Some labeling to point out the key features of the photo would help the audience understand the photo better.

Reviewer 4 Report

The manuscript describes the mechanical properties of the GFRP laminates with and without viscoelastic layer to show the influence of the layer on the properties with application to designing of the construction of planning hulls. In the work, the Authors conducted several tests which were described in detail in section 2. In my opinion, the results are new and interesting and a lot of solid experimental work was done. The results are presented in the correct form and are worthy to be published. The conclusions are supported by the results. As a general, I can recommend the work to be published after minor revision.

The biggest problem of the manuscript is an analysis of the experimental uncertainty. What is the uncertainty of the obtained results? Were the tests conducted only on one sample or more? The errors should be included in all plots and tables.

Round 2

Reviewer 3 Report

In this revision, the authors clarified some doubts I had with the initial version of the manuscript. One key information is the novelty of the work. The authors point out that the novelty of the current work, compared with their previous papers, is that an additional test condition involving the point-load impact on the composite is discussed.

The introduction of the paper is framed in a way that rationalizes the slamming test of the material, which is representative of the repeated pressure exerted by water waves on ships. However, given that the novelty of the work is really the point-wise drop impact test of the materials, the authors should re-frame the introduction to justify why this type of test is useful. For example, the paragraphs starting with line 50 and line 74 focus on slamming which is how the current experiment designed. Specifically, on line 52, the authors state: “The purpose of the investigation is to reduce the damage produced on composite material by slamming impacts”; however, this very same problem has been investigated in their previous papers and they have reached definitive conclusions (see P. Townsend et al. Ocean Engineering 159 (2018) 253-267; P. Townsend et al. Key Engineering Materials 847 (2020) 3-8). Likewise, on line 75, the authors raise the question: “Are they protected from the cyclical 75 impact of slamming according to the benefits and shortcomings of those presented in previous 76 studies?” Again, we already know the answer from their previous publications.

In contract, there is little information associate with vertical drop impact. It is unclear why is new test is essential for the design of the composite. The authors need to explain under which conditions would a ship experience a point-wise impact from water waves. Simply claiming that “aims to show another type of mechanical response of the viscoelastic layer” so that “the builders have different options for its use”, is too vague and does not justify the design of the experiment.

Compared with the previous work on slamming tests in Ocean Engineering 159 (2018) 253-267, what is new in the current test? It seems that the test materials and test conditions are very similar, with a similar sample size and a similar slamming pressure simulating the same conditions at sea. A similar FEM model was also presented therein.

For the slamming tests on unmodified panels, the six test cases specified in Table 1 of the current paper is strikingly similar to Table 2 of P. Townsend et al. Ocean Engineering 159 (2018) 253-267. The test results for cases D and E are the same with the Ocean Engineering (2018) paper. Regardless of whether the test data in the current manuscript is new or not, I do not see the rationale of publishing the same tests and results again.

Author Response

Dear Reviewer

For clarity, I have modified the introduction from line 106 to 116. In these lines it is explained that this work attempts to resolve new questions about viscoelastics that are completely linked to previous studies. I have included the Ocean Engineering reference so that readers understand the insertion of some data. It is necessary to include these data in the tables, because the purpose of the article is to clarify this benefit-problem and make it better understood.

The present work is an extension of the previous investigations of the authors of this paper, referring to the use of the viscoelastic layer to improve the performance in the GFRP. The tests carried out in previous studies are presented in this article with complements that try to solve the new questions formulated by the designers: is it feasible to install the viscoelastic sheets in the GFRP planing vessel? Are they protected from the cyclical impact of slamming according to the previously resolved benefits and shortcomings? [13-14] For this reason, the cyclical slamming tests are complemented with the vertical impact due to weight drop to present its benefits from the perspective that the designer wishes to observe.  The research aims to show another type of mechanical response of the viscoelastic layer under the perspective of vertical impact, and that the builders have different options for its use. The sheets are inexpensive and easy to cure in the construction of planing hulls.

Best Regards

Patrick Townsend
